# Prediction of Influenza Complications: Development and Validation of a Machine Learning Prediction Model to Improve and Expand the Identification of Vaccine-Hesitant Patients at Risk of Severe Influenza Complications

**DOI:** 10.3390/jcm11154342

**Published:** 2022-07-26

**Authors:** Donna M. Wolk, Alon Lanyado, Ann Marie Tice, Maheen Shermohammed, Yaron Kinar, Amir Goren, Christopher F. Chabris, Michelle N. Meyer, Avi Shoshan, Vida Abedi

**Affiliations:** 1Department of Laboratory Medicine, Diagnostic Medicine Institute, Geisinger, Danville, PA 17822, USA; amtice@geisinger.edu; 2Geisinger Commonwealth School of Medicine, Scranton, PA 18509, USA; 3Medial EarlySign, 6 Hangar Road, Hod Hasharon 4527703, Israel; alon@earlysign.com (A.L.); yaron@earlysign.com (Y.K.); avi@earlysign.com (A.S.); 4Behavioral Insights Team, Steele Institute for Health Innovation, Geisinger, Danville, PA 17822, USA; maheen.shermohammed@gmail.com (M.S.); agoren@geisinger.edu (A.G.); cfchabris@geisinger.edu (C.F.C.); mmeyer@geisinger.edu (M.N.M.); 5Department of Public Health Sciences, Penn State College of Medicine, Hershey, PA 17033, USA; vidaabedi@gmail.com

**Keywords:** electronic medical records, EHR, precision medicine, machine learning, decision support, influenza, risk stratification, vaccine, RT-PCR, Clinical Lab 2.0

## Abstract

Influenza vaccinations are recommended for high-risk individuals, but few population-based strategies exist to identify individual risks. Patient-level data from unvaccinated individuals, stratified into retrospective cases (n = 111,022) and controls (n = 2,207,714), informed a machine learning model designed to create an influenza risk score; the model was called the Geisinger Flu-Complications Flag (GFlu-CxFlag). The flag was created and validated on a cohort of 604,389 unique individuals. Risk scores were generated for influenza cases; the complication rate for individuals without influenza was estimated to adjust for unrelated complications. Shapley values were used to examine the model’s correctness and demonstrate its dependence on different features. Bias was assessed for race and sex. Inverse propensity weighting was used in the derivation stage to correct for biases. The GFlu-CxFlag model was compared to the pre-existing Medial EarlySign Flu Algomarker and existing risk guidelines that describe high-risk patients who would benefit from influenza vaccination. The GFlu-CxFlag outperformed other traditional risk-based models; the area under curve (AUC) was 0.786 [0.783–0.789], compared with 0.694 [0.690–0.698] (*p*-value < 0.00001). The presence of acute and chronic respiratory diseases, age, and previous emergency department visits contributed most to the GFlu-CxFlag model’s prediction. When higher numerical scores were assigned to more severe complications, the GFlu-CxFlag AUC increased to 0.828 [0.823–0.833], with excellent discrimination in the final model used to perform the risk stratification of the population. The GFlu-CxFlag can better identify high-risk individuals than existing models based on vaccination guidelines, thus creating a population-based risk stratification for individual risk assessment and deployment in vaccine hesitancy reduction programs in our health system.

## 1. Introduction

The human influenza virus causes substantial morbidity and mortality, often reducing the quality of life [1,2,3,4]; outbreaks have attack rates of 10–20 percent, but rates can exceed 50 percent in pandemics [4,5]. Most influenza epidemics disproportionately affect the elderly [6], but a shift in the age distribution can occur during pandemics [7] or in association with comorbid conditions [4,7]. Influenza-associated complications highlight the need for improved vaccination efforts for all age groups [6]. Furthermore, healthcare organizations experience influenza-related increases in emergency department (ED) utilization, [8] economic burden [9], and antimicrobial over-utilization [10] during influenza season.

Influenza vaccinations are recommended for high-risk individuals [11], but few population-based strategies exist to identify those at the highest risk. Although vaccination efficacy varies, depending on the match between the vaccines developed and the circulating strains of influenza [3,11], influenza’s human and organizational burdens are mostly preventable. Unfortunately, individuals often exhibit vaccine hesitancy or are unaware of their risks; therefore, they do not avail themselves of influenza vaccination [11]. These individuals can experience severe consequences and are sometimes over-utilizers of healthcare and emergency care [8].

The Centers for Disease Control and Prevention (CDC) and the World Health Organization (WHO) provide epidemiology and surveillance results and list risk factors related to age, health conditions, race, and congregate living conditions [12,13]. The published evidence describes vulnerable populations at risk for influenza complications by traditional methods unrelated to machine learning [3,9,11,14,15,16,17,18]. One study created a risk score for intensive care patients with influenza [19]. Another created clinical prediction rules by using artificial intelligence to analyze data from telemedicine visits for patients who could be infected by influenza [20]. No studies have taken an expansive population-health approach to creating individualized influenza complications risk scores.

Influenza-associated mortality estimates vary between studies due to differences in study settings, methods, and outcome measurements [21], confounding systematic comparisons. In a WHO systematic review performed [21], no “average” estimate of excess mortality was made due to the substantial variability of the mortality estimates. Global influenza risk factors are assessed periodically [18], but composite influenza risk stratification is generally limited to age and a few specific high-risk populations [22,23,24,25].

To date, there is no standard method, machine learning or otherwise, to assess an individual’s risk for influenza complications. Likewise, there is no definitive method to perform risk stratification on an entire population; therefore, risk stratification is rarely pursued [26]. Similarly, precision medicine strategies to rapidly treat influenza infection based on precise, rapid test results to prevent long-term complications do not exist. Reproducible population-based approaches using individualized risk profiles or personalized severity scores might help target vaccine hesitancy by informing patients of their high risk of infection and complications.

Gaps in the predictive modeling literature include a lack of inclusion of laboratory data and testing trends; accurate detection of influenza infection by molecular methods; and the limited ability to assess the multifactorial impacts of smoking, socioeconomic status (SES), previous ED visits, medications, history of acute respiratory illness, peripheral capillary oxygen saturation (SPO_2_), vital signs, or sex. Nevertheless, there is a possibility that relative standardization can occur among a single healthcare system or across harmonized systems and subsequently identify individuals with the highest risk of post-influenza sequelae or death.

The current project aims to develop a population-based machine learning (ML) tool to identify individuals at the highest risk of developing severe influenza infections and complications by uncovering unique risk attributes. Potential race and sex biases in the ML algorithm are assessed. Inverse propensity weighting is used in the derivation stage to correct for biases. The goal is to use the ML risk stratification system to drive a cost-effective approach to improve influenza vaccination in high-risk individuals, identifying those most likely to experience extreme complications for a personalized follow-up to communicate their risks.

## 2. Materials and Methods

### 2.1. Aim

This study aimed to develop and validate machine learning (ML) models to identify unvaccinated high-risk individuals, predicting the probability of acquiring influenza and developing influenza-related complications.

### 2.2. Population and Setting

This study was performed at Geisinger (a multi-hospital system in Central and Northeast PA, USA) in collaboration with Medial EarlySign (Hod Hasharon, Israel). The data originated from a de-identified data lake of >641,000 unique individuals who received Geisinger primary care services from 1 October 2008 to 31 January 2018 (i.e., the membership period) when vaccination coverage was 32.9–36.7%. After filtering individuals without longitudinal data, the final cohort consisted of unique unvaccinated individuals, representing 2,318,736 patient years, with influenza and one or more complication(s) within three months or none for at least nine months after infection (n = 604,389).

### 2.3. Definitions and Registries

Appendix A) lists the model and time-window features. An influenza season was defined to begin on 1 September and end on 1 May. The complication follow-up continued until 31 July (Figure 1). Influenza events defined the registries (Figure 2). Cohort membership was based on outpatient encounters. Exclusion criteria and cohorts used for model testing were determined (Figure 2A).

To mitigate diagnosis inaccuracy, two confidence levels defined two corresponding influenza registries within the cohort (Figure 2B). The Laboratory Test Registry (LabReg) used positive laboratory tests for influenza diagnosis, Appendix A. The more broadly defined Phenomic Registry (PheReg) used influenza-like illness (ILI), defined by ICD codes or Tamiflu usage, Appendix A. 

### 2.4. Data Pre-Processing

Geisinger stores ICD codes within internal (EDG) codes in Epic software (Madison, WI, USA). For the study, Geisinger EDG and ICD-9 codes were converted to ICD-10 codes (Appendix A).

### 2.5. Severity Tiers

Once placed into a registry, influenza complications were categorized into three severity tiers: death, hospitalization (in-patient or ED visits), and severe illness (e.g., pneumonia) (Appendix A and Figure 2).

### 2.6. Probability Characterization and Performance Measure Calculation

Influenza cases with non-influenza-related comorbidities were determined to define post-influenza complications properly; probability equations categorizing individuals before model training and validation are listed (Figure 3). “True” cases were defined as complications with a preceding influenza event, Equation (1). “Observed” cases were defined as a complication after an influenza event, regardless of possible causation (either “true” cases or random temporal positioning of influenza and non-related complications), Equation (2), and estimated by the product of two equations: Equation (3), for estimating the true case probability from observed, and Equation (4), counting the unrelated complications minus observed influenza cases followed by complications.

### 2.7. Model Training, Testing, and Validation

The GFlu-CxFlag model was trained on Geisinger’s dataset; training and test samples were generated. Each individual was randomly assigned to an ML subset: 70% was assigned to the training subset, 20% to the test subset for model testing, and 10% was saved for model validation.

### 2.8. Feature Generation and Selection

A set of categorical features was generated for each sample (e.g., ICD-10 codes, anatomical therapeutic chemical codes (ATCs), hospital admissions and transfers, and current procedural terminology (CPT) codes). Multiple time-window-dependent features were generated for each category and several time windows to create intuitive and explainable features (e.g., pneumonia events over the last five years). The approach (Appendix A) resulted in an extensive matrix with 698,780 features. The ICD-10 features’ hierarchies were examined using algorithmic logic and clinical intuition, Appendix A, to choose between descendants and ascendants if both showed significant dependence.

### 2.9. Model Development

The classifier used was XGBoost [26], an algorithm from the Gradient Boosting Machines family; it performed better than logistic regression. Model development and tuning used 6-fold cross-validation to maximize the AUC when testing on unvaccinated individuals to avoid overfitting. The optimization process tested XGBoost parameters with several training and weighting options on trained samples, with and without vaccinated individuals (Appendix A). Blinded validation occurred with subjects randomly placed into ML subsets. For parameter tuning, 156 runs were performed within the MES ML software environment. Appendix A lists the XGBoost parameter tests and results. A weighting process was used during model training, Figure 3 Equation (5), to correct for unrelated complications.

After pre-processing and data modeling, two models were selected for final development: GFlu-CxFlag, a “full” model using 147 data features, including vital signs, laboratory results, and clinical procedures, and by applying iterative backward feature selection, a smaller set of features was used to create the MES Flu Algomarker (Appendix A).

### 2.10. Model Evaluation

The final models were compared with the simpler CDC/WHO risk assessments converted to ML models. Bootstrapping was used to estimate confidence intervals and standard errors of performance measurements. Performance was compared using an XGBoost model trained with age and sex, in addition to age, sex, and comorbidities.

### 2.11. Propensity Analysis for Predicting Potential Vaccination

Because the GFlu-CxFlag model was trained on unvaccinated individuals, inverse propensity weighting (IPW) in the MES environment was used to validate the model and adjust for population bias; it was not used in calculating risk scores. For IPW analysis, the model was trained to predict whether individuals would get vaccinated using historical patient communications (Table 1).

### 2.12. Bias Assessment

Model bias was evaluated with four sociodemographic characteristics: race, ethnicity, sex, and socioeconomic status (SES); Medicaid insurance was a surrogate for low SES. Sensitivity across different characteristic categories was compared; chi-squared tests determined statistical significance, with two-tailed *p* < 0.05 criteria defined to identify potential evidence of bias. A reference group, to which all other categories were compared in a pairwise fashion, was chosen for characteristics with more than two categories: White for race; Medicaid for SES.

To probe for possible bias sources across groups exhibiting model biases, random sampling created sub-groups that were matched on dimensions for which model performance was expected to vary: age and amount of data (defined as visits/last five years). Sensitivity was re-evaluated using these matched sub-groups. We applied the same process to a “model” that used a simple age cutoff to classify individuals > 65 years of age as “high risk” as a means to contextualize bias. Appendix A depicts sensitivity for individuals categorized by each attribute of interest.

## 3. Results

### 3.1. Data Features

The data contained about 590,000 individuals/year. The case distribution/year exhibited high variability due to varying influenza severity. The monthly distribution fits seasonal patterns, peaking in January. The LabReg included 25,156 events/10 years (0.5–1% each year). The PheReg contained 1,300,045 events/10 years (12.1–17.6%). There were more events for young and elderly individuals each year.

Adjusting for non-influenza-related complications reduced the influenza complications’ case count by approximately 22%, indicating that certain post-influenza complications occurred within three months, even without preceding influenza infection(s). After filtering and matching for the influenza season, the training set, > 1.6 million data points, had 2371 features for the GFlu-CxFlag, 334 for the MES Flu Algomarker, and 15 for CDC/WHO model. Most laboratory features did not contribute significantly to model performance and were eliminated from the MES Flu Algomarker. The addition of the lymphocyte percentage feature slightly improved the full model performance, as did respiratory rate and SPO_2_.

### 3.2. GFlu-CxFlag Performance

#### 3.2.1. GFlu-CxFlag Comparison to Other Models

Table 2 depicts the GFlu-CxFlag performance for several cutoff scores. The AUC of 0.790 [0.780–0.790] was documented for all populations and outperformed other models. Table 3 depicts the AUC performance, subcategorized by test set sub-populations, representing the discriminative performance of all models on unvaccinated individuals, substratified by age, without applying IPW or correction for over-estimation due to unrelated complications. When the IPW correction was applied, the AUC was 0.786 [0.783–0.789]. The model performance on the LabReg improved the performance. The MES Flu Algomarker AUC was 0.783 [0.780–0.787]. The CDC/WHO model AUC was 0.694 [0.690–0.698].

The GFlu-CxFlag model significantly outperformed the CDC/WHO model (*p* < 0.00001), identifying unique features (Table 4), when a 5% false-positive rate was assigned as the cutoff; other respiratory diseases, age, and previous ED admission contributed most to prediction. The performance for training on both vaccinated and unvaccinated individuals was less robust, even when testing occurred in the cohort containing unvaccinated and vaccinated individuals. The training process weighting method improved the model performance slightly in all analyses, even when measuring AUC without corrections or not using IPW on unvaccinated individuals.

#### 3.2.2. GFlu-CxFlag, Comparisons When Substratified by Severe Complications

To support the claim that GFlu-CxFlag ranks more severe influenza complications higher, the model discrimination between influenza complications cases was tested by severity tiers 1 and 2, without 3. The cohort was changed to include only individuals who experienced influenza complications (n = 22,116). When the least severe complications (tier 3) were labeled as controls and severity tiers 1 and 2 were labeled as cases, the AUC was 0.596 [0.586–0.606], confirming the model ranked the more severe cases higher. The mean risk-severity score for tiers 1 and 2 was 0.160 [0.156–0.163] with 9648 samples compared to 0.119 [0.117– 0.122] with 12,468 samples for tier 3 (*p* < 0.00001).

The propensity model performance for the GFlu-CxFlag IPW correction reached an AUC of 0.869 [0.868–0.870]. The most important features were vaccination, age, gender, and clinical characteristics, such as influenza, vaccination history, smoking, hyperlipidemia, temperature, weight, psycho-analeptic drugs, and lipid-modifying agents.

#### 3.2.3. Evaluating Feature Contributions to GFlu-CxFlag

Figure 4 shows the feature contribution, ordered by the mean absolute Shapley values. The top four contributing features linked the history of respiratory-related and general comorbidities. The most important category was ICD10:J00-J99—a superset of respiratory diseases, followed by years of data, complications, and influenza history. The temporal membership features documented data missingness, important for features that use time windows, and allowed for normalization of numerical features, such as the number of ED visits, substratified by the time period in which they were counted.

Figure 5 shows model and data behavior as functions of the important features. The *x*-axis represents the feature value, and the yellow lines represent the mean outcome over the training set conditioned on the feature value. The blue line represents the feature’s mean Shapley value. The average score, conditioned on feature value, was similar to the mean outcome (data not shown) in all cases. As depicted in Figure 5A, the U shape was expected for the contribution of age; very young and very old individuals have a higher risk of complications. Figure 5B shows that the complication risk increased with the number of respiratory diseases over the last five years, defined by the history of ICD10:J00-J99. The complication risk decreased as time since smoking cessation increased (Figure 5C). Increased risk in individuals who quit smoking long ago (a small set) is not reflected in the Shapley value, indicating that the model did not overfit. Instead, the model attributed the higher risk to old age (e.g., 80 years old since quitting means the individual was old). Figure 5D shows a U-shaped behavior in the mean outcome as a function of body mass index (BMI)—a young age is a likely confounder associated with a lower BMI. A high BMI was an independent risk factor, reflected in the mean Shapley values, which remained low at a low BMI, but monotonically increased with a higher BMI.

#### 3.2.4. Post-Processing (GFlu-Cx Flag Bias Assessment)

Post hoc analysis is depicted in Table 5 and Appendix A. For race, Gflu-CxFlag revealed significantly higher sensitivity for White than for Black individuals (X^2^ = 7.4, *p* = 0.006), suggesting an algorithmic bias favoring White individuals. The difference was ameliorated after age-matching (White: 41.6 [41.1–42.1], Black: 40.3 [38.2–42.6], X^2^ = 1.83, *p* = 0.176), suggesting that age differences between groups may drive bias. A simplistic “model” tagging anyone > 65 years old, as high risk would produce a stronger White-favoring bias (X^2^ = 123.56, *p* < 0.001). There was a significant difference in model sensitivity between White and Asian individuals (X^2^ = 7.89, *p* = 0.005); this difference decreased but remained significant after age-matching (White: 40.0 [39.1–41.1], Asian: 26.4 [15.2–38.1], X^2^ = 6.07, *p* = 0.014).

For ethnicity, there was a significant difference in model sensitivity favoring Hispanic/Latin American individuals (X^2^ = 7.11, *p* = 0.008), which was mitigated by age-matching (Hispanic: 41.4 [39.3–43.6], non-Hispanic: 41.2 [40.7–41.7], X^2^ = 0.04, *p* = 0.848). For low SES (current or within the previous 11 years), the model revealed significantly greater sensitivity for individuals on Medicare than recently on Medicaid (X^2^ = 7.29, *p* = 0.007) and greater sensitivity for individuals recently on Medicaid than under commercial insurance (X^2^ = 818.63, *p* < 0.001). The Medicare vs. Medicaid effect was reversed after age-matching (Medicare: 54.1 [52.4–55.7], Medicaid: 59.2 [57.7–60.7], X^2^ = 23.23, *p* < 0.001), but the Medicaid vs. commercial effect remained (Medicaid: 50.3 [49.5–51.3], commercial: 28.1 [27.4–28.7], X^2^ = 2111.2, *p* < 0.001), continuing to exhibit bias favoring the more vulnerable group in this category.

For sex, the model revealed greater sensitivity for female than for male individuals (X^2^ = 61.54, *p* < 0.001). This effect remained after age-matching (female: 45.8 [45.2–46.3], male: 40.1 [39.5–40.7], X^2^ = 216.92, *p* < 0.001), but was mitigated after matching for age and the number of visits available (female: 37.7 [37.0–38.4], male: 37.7 [37.0–38.4], X^2^ = 0, *p* = 1).

## 4. Discussion

Human and healthcare influenza burden remains high [18]; therefore, a process to improve risk-stratification was created. GFlu-CxFlag improved sensitivity for identifying unvaccinated individuals with the highest risk for influenza and complications compared with the CDC/WHO model by 86% when a 5% false-positive rate was the cutoff. The improvement will identify an additional 33.1% of influenza complications compared with 17.8% with the CDC/WHO model used with Geisinger data. GFlu-CxFlag is generalizable to other data-rich organizations; the MES Flu Algomarker and the CDC/WHO model could be implemented using most current electronic health software programs.

The bias analysis did not reveal any significant biases against Black, Hispanic, or Latin American individuals; Medicaid patients; or females, which could not be accounted for by differences in predictive features, such as age or number of visits. For Black individuals, subpopulation differences in age appear to account adequately for the lower observed sensitivity, suggesting that individuals of the same age as their White counterparts should be flagged as being at the same risk as identified by other predictors. GFlu-CxFlag use may be more limited for Black individuals when compared with White individuals; however, the model results in an almost threefold improvement in performance for this group when pragmatically compared with the typical age-based risk-stratification method. Similarly, insurance coverage disparities between Medicare and Medicaid are significantly reduced when accounting for age, suggesting the model may not be biased against poorer populations and favors these individuals in some cases. The bias evaluation indicates the model is appropriate and highlights steps to identify sources of bias and make future model adjustments.

Geisinger’s data-rich environment is a study advantage due to population longevity and the low percentage of geographic movement. Limitations may include a high insurance coverage rate for individuals, including healthcare employees (commercial insurance coverage 48.5%, 36.1% Medicaid, and 14.5% Medicare).

Due to biases in the underlying data or the social processes that generate them, ML algorithms can propagate or exacerbate biases against under-represented groups traditionally facing discrimination. After accounting for age, bias remains against one group: Asian individuals (N = 87); results should be interpreted with caution.

GFlu-CxFlag was impacted by inaccurate ILI documentation since it encapsulates both general fragility risk and the probability of ILI, which is challenged by medical coding heterogeneity. The impact of accurate test results is difficult to disentangle. Due to the model’s elimination process, many different solutions occur. “Richer” more common information sources, such as diagnosis codes and medications, are important for broad inclusion; therefore, more specific laboratory tests were saved until the end of the elimination process, the likely reason for the small, redundant impact. Future study of the variable elimination “order” could lead to a more comprehensive model understanding.

The RT-PCR impact cannot be discounted because the effect was absorbed into the diagnosis and complications of influenza, thereby “flowing” through other data sources. RT-PCR counts were lower in the early years, minimizing test impact by approximately 30%. Based on the higher AUC of the LabReg, an accurate identification of influenza could continue to improve model prediction in the future.

Despite the promising results, the model must perform well over time and in other organizations. Users who do not use the MES ML environment would need to recreate models with their data. Several population-based models, including Google’s Flu Trends [19], attempted to describe the general severity of influenza seasons. Nonetheless, there is disagreement on how helpful predictive modeling is and what benefit it serves for a healthcare community (https://time.com/23782/google-flu-trends-big-data-problems, accessed on 1 July 2022). If GFlu-CxFlag was applied prospectively, seasonal variables would need to be estimated.

The Geisinger Flu-Complications Flag (GFlu-CxFlag), created in conjunction with Medial Early Sign (MES), uses many more conditions than other models. According to 2020 population data, the improvement reflects the identification of nearly 641,000 unique individuals in the entire primary care population of the health system, serving a catchment area of approximately three million people in a rural region of the United States. The 10% at highest risk for influenza complications were identified as high risk. Extrapolated to the US, 10% recognitions could be over 33 million high-risk individuals, and globally 770 million. Healthcare systems could adapt the model to target vaccination outreach more effectively than using age, sex, and comorbidity cutoffs alone. Because different healthcare systems may not capture the same variables used in this study, the value of the study can still help identify some core model parameters in other centers. Finally, this work has implications for identifying risk factors for COVID-19 to advance the prediction of the first version of the MES COVID Complications AlgoMarker.

## 5. Conclusions

The GFlu-CxFlag is a significant new contribution to risk-stratification strategies, supporting more accurate risk calculation for influenza-related morbidity and mortality by identifying key factors contributing to severe complications in different sub-groups of individuals. Using a GFlu-CxFlag-like approach, healthcare organizations could combine their risk-stratification and vaccination efforts to advance vaccine uptake.

The findings add to the scientific literature that may help mitigate the impact of vaccine hesitancy. Current vaccine recommendations from the World Health Organization (WHO), the USA Center for Disease Control and Prevention (CDC), and the Israeli Ministry of Health (MOH) recommend vaccination for the entire population at six months of age and older, with an emphasis on the importance of vaccination for people at a higher risk of severe influenza complications. According to the CDC, high-risk groups include individuals with long-term diseases, such as acquired or congenital cardiovascular disease, congestive heart failure, atherosclerosis, diabetes, and other chronic metabolic diseases; chronic diseases. Chronic illness include chronic lung diseases, including asthma; chronic liver disease, chronic kidney disease and urinary tract infections; neurological and hematological diseases; and diseases accompanied by immunosuppression, including AIDS and malignant diseases. Additional special high-risk populations are pregnant and post-partum women, children aged 6 months to 6 years (and especially up to the age of 2 years), children aged 6 months up to 18 years that receive long-term aspirin therapy, and individuals 50 years old and above, especially 65 and above. The WHO further identifies pregnant women as the highest risk priority. This study uses primary care data and the machine learning modeling to improve the CDC/WHO guidelines for predicting the risk of future morbidity and mortality from influenza infections by 86%.

Our machine learning (ML) approach to risk stratification provides an essential new contribution to the field by determining the baseline rates of morbidity and mortality that reflect conditions other than age, sex, and limited comorbidities. The approach allows for a more accurate calculation of influenza-related morbidity and mortality, which could be generalizable to influenza vaccine campaigns and provide helpful information to policymakers. Future research can use these tools and strategies to understand vaccine campaigns for COVID-19. Adopting the GFlu-CxFlag could expand the identification of high-risk individuals, reducing influenza’s human and organizational impact. If the GFlu-CxFlag was adopted for predicting influenza-associated complications, the results would translate to the identification of approximately 64,000 high-risk individuals in a Geisinger-like system serving a catchment area of roughly three million individuals. Extrapolated to the US, the prediction could reach 33 million and 770 billion globally.

## Figures and Tables

**Figure 1 jcm-11-04342-f001:**
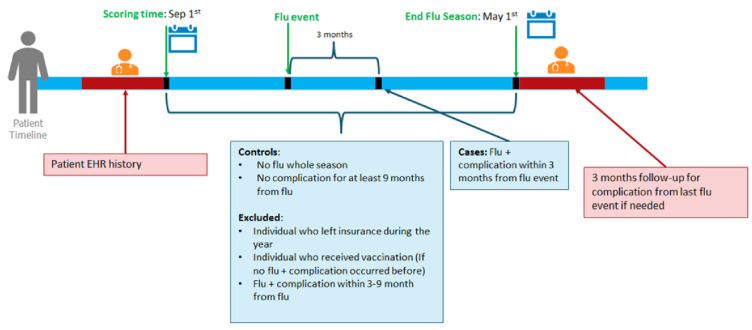
Inclusion and exclusion criteria with definitions of cases and controls used during data pre-processing: Description of cases, controls, and exclusion criteria during data pre-processing, i.e., cohort definition of influenza-related complications for unvaccinated individuals within a given influenza season.

**Figure 2 jcm-11-04342-f002:**
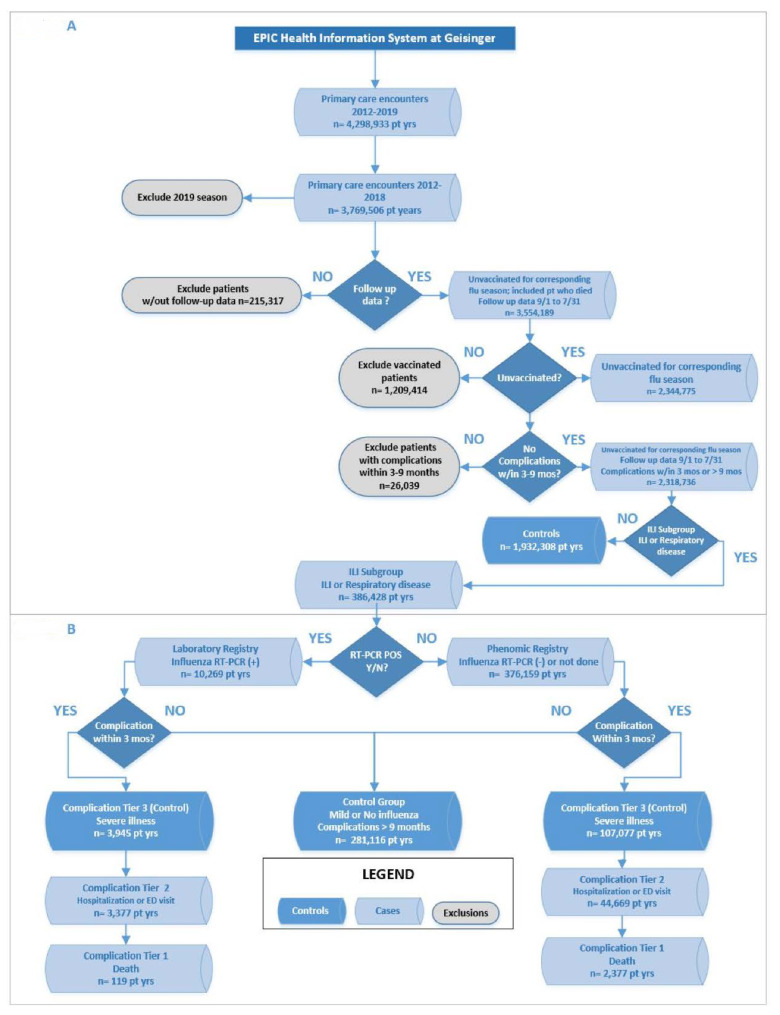
Tiers of confidence for influenza diagnosis and severity levels for influenza complications: An individual could be included in the cohort several times for different influenza seasons but was unique within an influenza season. The same individual could be categorized as a case in one season and as a control in another. (**A**) represents the exlusion criteria and cohorts. (**B**) represents the influenza registries.

**Figure 3 jcm-11-04342-f003:**
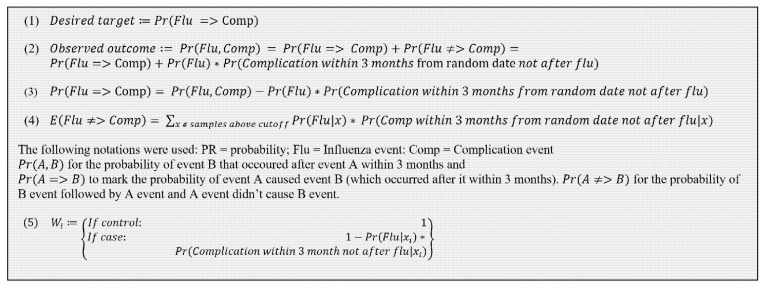
Equations Used. Equations (1)–(4) were used for defining the probability for complications. Equation (5), which describes the weighting process used in the model training stage, where Xi is the data vector for sample *i* and *Wi* is the weight for sample training *i*.

**Figure 4 jcm-11-04342-f004:**
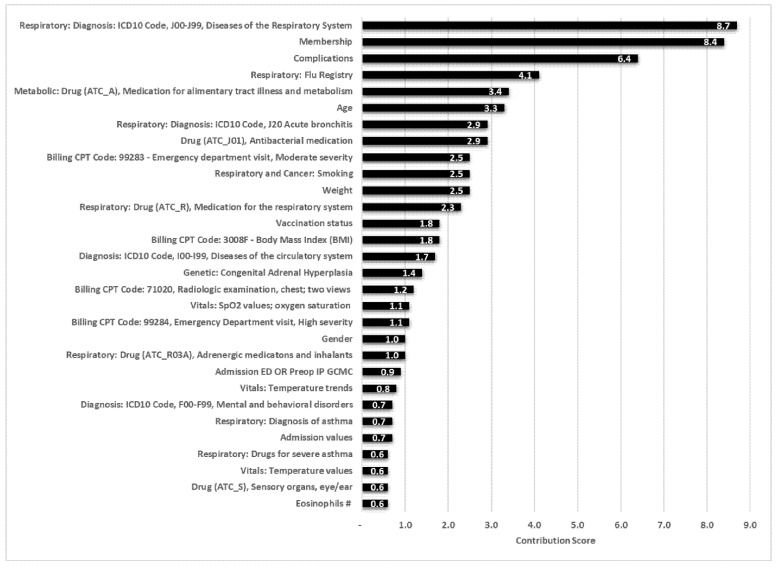
Influenza complications for the model’s global feature importance (Geisinger Flu-Cx Flag). Refer to Appendix A for description of codes used in Figure 4.

**Figure 5 jcm-11-04342-f005:**
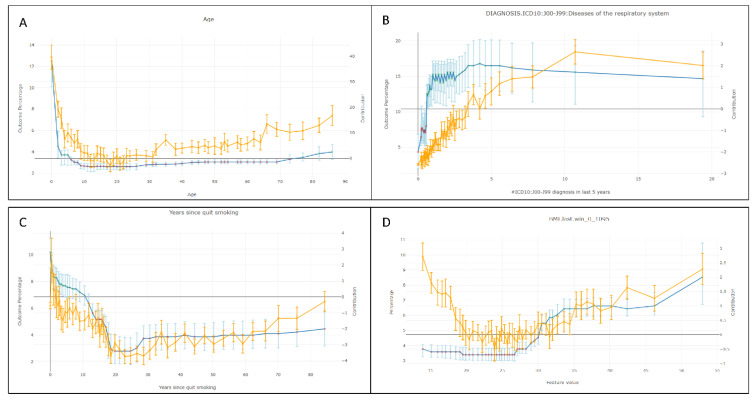
Feature contribution for various features in the full Geisinger Flu-Cx Flag model. The *x*-axis represents the feature value, and the yellow lines represent the mean outcome over the training set conditioned on the feature value. The blue line represents the feature’s mean Shapley value.

**Table 1 jcm-11-04342-t001:** Comparison of AUC by different analysis methods and outcome definitions for different models.

	Age, Sex Model	WHO-inspired Age, Sex, and Comorbidities Model	Full GFlu-CxFlag Model
No IPW, no over estimation analysis	0.588 [0.583–0.593]	0.694 [0.690–0.698]	0.786 [0.783–0.789]
Only IPW	0.597 [0.592–0.602]	0.715 [0.711–0.720]	0.789 [0.785–0.793]
IPW and over estimation analysis	0.587 [0.580–0.593]	0.693 [0.687–0.699]	0.761 [0.757–0.768]
Flu diagnosis by RT- PCR & all complications *	0.632 [0.615–0.647]	0.704 [0.688–0.720]	0.797 [0.785–0.809]
Severe complications (cohorts 1 and 2) *	0.610 [0.602–0.616]	0.709 [0.703–0.716]	0.828 [0.823–0.833]

* Full model performance is based on a model trained for influenza probability categories 1–3 and severity/complication tiers 1–3 (with no retraining); it is a single model. Simplest model = Age and sex only; CDC/WHO Model = Age, Sex, and some Comorbidities; GFlu-CxFlag Full model; IPW = Inverse probability weighting; RT-PCR = reverse transcriptase polymerase chain reaction.

**Table 2 jcm-11-04342-t002:** XGBoost model discrimination and performance comparison table.

	Age, Sex Model	WHO Inspired (Age, Sex and Comorbidities Model)	Full GFlu-CxFlag Model
AUC	0.59 [0.58–0.59]	0.69 [0.690–0.70]	0.79 [0.78–0.79]
N, Controls	442,329 [441,033–443,687]	442,329 [441,022–443,739]	442,329 [440,907–443,499]
N, Cases	22,116 [21,747–22,500]	22,116 [21,775–22,428]	22,116 [21,773–22,473]
PPV@SENS_10	9.12 [8.62–9.63]	15.82 [15.27–16.42]	43.06 [41.23–44.93]
PPV@SENS_20	7.52 [7.27–7.80]	14.80 [14.21–15.40]	33.53 [32.40–34.89]
PPV@SENS_30	6.93 [6.73–7.16]	13.27 [12.90–13.70]	26.67 [25.67–27.73]
PPV@SENS_40	6.46 [6.30–6.64]	12.19 [11.78–12.54]	21.55 [20.88–22.25]
PPV@SENS_50	6.10 [5.97–6.26]	10.58 [10.26–10.89]	17.60 [17.03–18.21]
PPV@SENS_60	5.91 [5.79–6.04]	9.10 [8.86–9.37]	14.31 [13.90–14.74]
PPV@SENS_70	5.69 [5.58–5.81]	7.79 [7.61–8.01]	11.47 [11.13–11.84]
SENS@FPR_01	3.24 [3.01–3.49]	4.31 [4.04–4.61]	13.12 [12.55–13.71]
SENS@FPR_05	10.02 [9.59–10.49]	17.76 [17.14–18.38]	33.10 [32.44–33.82]
SENS@FPR_10	16.83 [16.28–17.41]	30.54 [29.82–31.36]	46.69 [46.01–47.42]
SENS@FPR_15	23.46 [22.84–24.14]	40.94 [40.21–41.67]	55.66 [54.97–56.37]
SENS@FPR_20	29.83 [29.13–30.61]	48.58 [47.85–49.34]	62.73 [62.06–63.45]
SENS@FPR_30	41.13 [40.30–41.97]	60.06 [59.39–60.78]	72.62 [71.99–73.29]
SENS@FPR_40	51.74 [50.95–52.56]	68.89 [68.22–69.57]	79.79 [79.21–80.38]
SENS@FPR_50	62.33 [61.46–63.04]	75.97 [75.33–76.62]	85.43 [84.95–85.94]

Simplest model = Age and sex only; CDC/WHO Model = Age, sex, and some comorbidities; GFlu-CxFlag model (Full Model) AUC = area under the curve; N= sample size; PPV = Positive predictive value; SENS = sensitivity of the model FPR = False positive rate.

**Table 3 jcm-11-04342-t003:** Comparison of AUC for different subpopulations.

	Age, Sex Model	WHO inspired (Age, Sex and Comorbidities Model)	Full GFlu-CxFlag Model
All	0.588 [0.583–0.593]	0.694 [0.690–0.698]	0.786 [0.783–0.789]
Age 18–65 years with chronic illness	0.553 [0.546–0.559]	0.671 [0.664–0.676]	0.775 [0.770–0.781]
Age 0–18 years.	0.617 [0.608–0.626]	0.705 [0.698–0.712]	0.792 [0.786–0.799]
Age 0–5 years.	0.594 [0.582–0.605]	0.680 [0.670–0.690]	0.777 [0.768–0.787]
Age > 18 years.	0.574 [0.568–0.579]	0.690 [0.684–0.695]	0.783 [0.779–0.787]
Age > 5 years.	0.573 [0.568–0.578]	0.689 [0.684–0.693]	0.783 [0.779–0.787]
Age 5–18 years.	0.563 [0.553–0.575]	0.686 [0.676–0.695]	0.782 [0.774–0.790]
Age > 65 years.	0.521 [0.510–0.532]	0.670 [0.660–0.680]	0.812 [0.805–0.819]
>18 year. with comorbidity or >65 year.	0.537 [0.531–0.542]	0.669 [0.663–0.674]	0.786 [0.781–0.791]

Simplest model = Age and sex only; CDC/WHO Model = Age, Sex, and some Comorbidities; GFlu-CxFlag model (Full Model).

**Table 4 jcm-11-04342-t004:** Features identified in the Geisinger and MES risk-stratification models compared with traditional WHO and CDC risk factors.

	Model	GFlu-Cx Flag	MES Flu	WHO	CDC
Laboratory Results	% Lymphocytes				
	Absolute eosinophils				
	Reverse-transcriptase PCR confirmation of influenza				
Vital Signs	Respiratory rate				
	SpO_2_ (peripheral capillary oxygen saturation)				
	Temperature (Fahrenheit)				
Medical History	Respiratory disease, not limited to lung (acute or chronic)			**Lung only**	**Lung only**
	Alimentary or metabolic diagnosis codes				
	Influenza complications (pneumonia, complications, death)				
	Incidence of influenza-like illness documented				
	Antibiotic prescriptions/medication for sensory organs (ear or eye)				
	Vaccinations				
Demographics	Age				
	Sex				
	Socioeconomic status (Medicare as a surrogate)				
	Weight and/or BMI				**Obesity**
	Smoking history				
	Membership in Geisinger cohort				
Healthcare Interactions	Number of Emergency Dept. (ED) visits ^#^				
	Number of hospital admissions				
	Other unique features for CDC and WHO as listed ^&^				
**LEGEND**					
	Light-gray indicates unique or shared presence features from the WHO and/or CDC guidelines for populations at high-risk of influenza				
	Dark gray-shade indicates the top 5 features of importance in Geisinger Flu Complications Flag (Gflu-Cx Flag)				
	Gray-shade describes other unique features in Geisinger Flu Complications Flag (Gflu-Cx Flag)				
	Black-shade = Laboratory testing with RT-PCR, was unique but not a model feature because it was a classifier to the influenza diagnosis				

^#^ longitudinal trends were used as a measure of the variable; RT-PCR is not a model feature, because it is a classifier to the influenza diagnosis; its importance is underscored by the model’s prediction when RT-PCR is used to define illness; ^&^ WHO unique features = Lung, heart, kidney, neurologic, liver, and blood disease, plus immunocompromised status, stroke, pregnancy, and work in healthcare and CDC unique features = The same as WHO features, plus aspirin therapy, long-term care, and race. CDC risks do not include healthcare workers.

**Table 5 jcm-11-04342-t005:** Post-hoc analysis for bias assessment.

Group (% of Total Population)		Sensitivity [95% CI]	Effect of Matching	Aged 65+ “Model” Sensitivity [95% CI]
**Race ^1^**				
	White (92.6%)	43.1 [42.4–43.9]	-	16.68 [16.1–17.4]
	Black (5.3%)	38.8 [35.3–42.3] *	Mitigated after matching for age	3.86 [2.6–5.2] *
	Asians (1%)	27.6 [16.7–39.8] *	Maintained after matching for age	9.27 [3.0–16.7]
**Ethnicity**				
	Hispanic / LA (5.4%)	46.5 [43.3–50.0]	-	3.22 [2.1–4.5]
	Non-Hispanic / LA (94%)	42.5 [41.7–43.3] *	Mitigated after matching for age	16.59 [15.9–17.2] *
**Insurance Type ^1^**				
	Medicaid ^2^ (36.1%)	49.5 [48.4–50.5]	-	2.79 [2.4–3.2]
	Medicare (14.5%)	52.0 [50.4–53.7] *	Reversed after matching for age	70.59 [68.8–72.4] *
	Commercial (48.5%)	28.1 [27.0–29.3] *	Maintained after matching for age	4.32 [3.7–5.0] *
**Sex**				
	Female (53.3%)	45.0 [44.2–45.9]	-	16.63 [15.8–17.5]
	Male (46.7%)	39.7 [38.5–41.0] *	Mitigated after matching for age and number of visits	15.12 [14.2–16.1]

* Significantly different at *p* < 0.05 compared to reference category, always listed first; CI = Confidence Interval; LA = Latin American ^1^ Other race and insurer categories exist but each compose less than 1% of the population; ^2^ Patients enrolled in Medicaid at any point in the last 11 years were placed in this category, even if they later shifted insurance (e.g., aged into Medicare).

## Data Availability

Geisinger and their patients own the data used for the project; they was collected from an existing data lake within the Geisinger data architecture, which contains individuals with a Geisinger PCP. The data can be shared with academic researchers with investigational support to fund the data transfer. Individual participant data and a data dictionary defining each field in the set will be available to others as follows: a de-identified copy of the data lake can be shared if the appropriate documentation and data-use agreement are on file on the publication date and for five years after. Contact the Geisinger Research Institute at irb@geisinger.edu to obtain a data-use agreement.

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
