# Peer review of "Prediction of Influenza Complications: Development and Validation of a Machine Learning Prediction Model to Improve and Expand the Identification of Vaccine-Hesitant Patients at Risk of Severe Influenza Complications"

_jcm, 2022, doi:10.3390/jcm11154342_

Round 1
Reviewer 1 Report
The manuscript entitled "prediction of influenza complications: Development and validation of a machine learning prediction model to improve and expand identification of vaccine-hesitant patients at risk for severe influenza complications" used machine learning algorithms to predict the risks of influenza complications in non-vaccinated individuals. I found the flow of this manuscript very difficult to follow, because:
1) The introduction is too short to provide a comprehensive overview of the background information.
2) The key features included in the ML model are not clearly stated. In the abstract, it shows that "overall eight unique features were identified", but I cannot find what exactly these 8 features are. In figure 4, there are more than 8 features.
3) Important information is put in the figures, but figures are too small to see. I cannot read the small font in the figures.
4)The AUC of the ML model (0.786) is not impressive. I would expect a higher AUC to be clinically useful.
5) Language editing is needed. Many sentences are confusing. For example, in the first paragraph of the result session, "there are more events for young and elderly individuals each year". What are the definition of "young" and "elderly"? Are you comparing them to other age groups? Please be specific.
Author Response
The introduction was expanded.
Additional references were added.
Additional detail was added to the text, including definitions of children and elderly in the Geisinger system.
Table 3 was modified. Other figures were modified to make the text size more readable.
We agree that the AUC of the model may be lower than the reviewer expected. We respectfully ask that the following aspects are considered: 1) the extensive sensitivity analysis, subgroup analysis, and extremely large cohort are part of the lower AUC, making it reasonable under the said conditions. When the data set is very small, the AUC can be falsely high, sometimes occurring due to selection bias and the fact that the small cohort can be very homogenous. According to AUC considerations, the following apply
- 0.5 = No discrimination
- 0.5-0.7 = Poor discrimination
- 0.7-0.8 = Acceptable discrimination
- 0.8-0.9= Excellent discrimination
- >0.9 = Outstanding discrimination
By these standards, a model with an AUC score below 0.7 would be considered poor, and anything higher would be considered acceptable or better.
The reported AUC of 0.786 borders on 0.8, close to the excellent discrimination mark. We also assert that this was not built to identify a clinical disease when one might require a higher AUC. The final model used had an AUC > 0.8 (excellent discrimination); 0.828 [0.823 – 0.833]. The model was built to create a risk- stratification to encourage people to accept influenza vaccination. The model statistically improved influenza vaccination in our system over the last 2 years - that publication is in progress. The model is doing its job for the intended purpose. Furthermore, the Geisinger model improves upon the traditional risk listing described by CDC and WHO when they were tested on the same data.
Finally, the manuscript was put through the process of Grammarly software before it was first submitted. Grammarly and Word grammar checkers, our intent was to optimize grammar for the general science reader. the wording may seem too simplistic compared to some scientific writing; however, the authors wrote the manuscript with a wide diversity of people in mind - clinicians, laboratorians, hospital administrators; employee health nursing; pharmacists, etc. The English grammar was adjusted to a general medical audience for that reason. We respectfully propose this is a style choice, not grammar inaccuracy; however, we await feedback on our re-submission with these factors in mind.
Reviewer 2 Report
1. What is GFlu-CxFlag model? What is the input and output? Which machine learning is used? More details should be given.
2. How is the significant dependence evaluated for feature generation and selection?
3. Line 124 “Propensity analysis for predicting potential vaccination” is not complete.
Author Response
the introduction was expanded and more key references were added.
The research design was better explained
More details was added to methods and tables, supporting the methods
For results, figures and tables were improved
Propensity analysis was not a fragment, it should have been a header and it's been revised as such. 2.11 Propensity analysis for predicting potential vaccination
The classifier used was XGBoost line 149 and Supp Table 6 as indicated below. The optimization process tested XGBoost parameters with several training and weighting options on trained samples, with and without vaccinated individuals, Supplementary Tables 6.